# The Role of Microglial Exosomes and miR-124-3p in Neuroinflammation and Neuronal Repair after Traumatic Brain Injury

**DOI:** 10.3390/life13091924

**Published:** 2023-09-16

**Authors:** Ioannis Mavroudis, Ioana-Miruna Balmus, Alin Ciobica, Mircea Nicusor Nicoara, Alina Costina Luca, Dragos Octavian Palade

**Affiliations:** 1Department of Neurology, Leeds Teaching Hospitals, NHS Trust, Leeds LS2 9JT, UK; ioannis.mavroudis@gmail.com; 2Faculty of Medicine, Leeds University, Leeds LS2 9JT, UK; 3Department of Exact Sciences and Natural Sciences, Institute of Interdisciplinary Research, Alexandru Ioan Cuza University of Iasi, Str. Alexandru Lapusneanu, no. 26, 700057 Iasi, Romania; 4Department of Biology, Faculty of Biology, Alexandru Ioan Cuza University of Iasi, Bd. Carol I no. 20A, 700505 Iasi, Romania; 5Centre of Biomedical Research, Romanian Academy, Bd. Carol I, no. 8, 700506 Iasi, Romania; 6Academy of Romanian Scientists, Str. Splaiul Independentei no. 54, Sector 5, 050094 Bucharest, Romania; 7Preclinical Department, Apollonia University, Păcurari Street 11, 700511 Iasi, Romania; 8Doctoral School of Geosciences, Faculty of Geography and Geology, Alexandru Ioan Cuza University of Iasi, Bd. Carol I no. 20A, 700505 Iasi, Romania; 9Faculty of Medicine, “Grigore T. Popa” University of Medicine and Pharmacy, Str. Universitatii no. 16, 700115 Iasi, Romania

**Keywords:** traumatic brain injury, mesenchymal stem cell-derived exosomes, microglia, miR-124-3p

## Abstract

(1) Background: In this study, we aimed to explore the regulatory mechanism of miR-124-3p microglial exosomes, as they were previously reported to modulate neuroinflammation and promote neuronal repair following traumatic brain injury (TBI). (2) Methods: Studies investigating the impact of microglial exosomal miRNAs, specifically miR-124-3p, on injured neurons and brain microvascular endothelial cells (BMVECs) in the context of TBI were reviewed. (3) Results: Animal models of TBI, in vitro cell culture experiments, RNA sequencing analysis, and functional assays were employed to elucidate the mechanisms underlying the effects of miR-124-3p-loaded exosomes on neuroinflammation and neuronal repair. Anti-inflammatory M2 polarization of microglia, mTOR signaling suppression, and BMVECs-mediated autophagy were reported as the main processes contributing to neuroprotection, reduced blood-brain barrier leakage, and improved neurologic outcomes in animal models of TBI. (4) Conclusions: Microglial exosomes, particularly those carrying miR-124-3p, have emerged as promising candidates for therapeutic interventions in TBI. These exosomes exhibit neuroprotective effects, attenuate neuroinflammation, and promote neuronal repair and plasticity. However, further research is required to fully elucidate the underlying mechanisms and optimize their delivery strategies for effective treatment in human TBI cases.

## 1. Introduction

Comprehensive evidence shows that these small extracellular vesicles play a vital role in intercellular communication by transporting a wide range of bioactive molecules, including proteins, nucleic acids, and lipids [1,2,3]. A growing interest in them as potential therapeutic targets arose. The process through which they are generated is based on a complex cellular pathway involving the double invagination of the plasma membrane and the formation of intracellular multivesicular bodies (MVBs) that consist of intraluminal vesicles (ILVs) [4,5,6,7,8,9]. Initially, the plasma membrane invaginates to form a cup-shaped structure that includes the cell surface and soluble proteins associated with the extracellular space. Afterwards, the formation of an early-sorting endosome (ESE) is performed by direct merging with a preexisting ESE in some cases while interacting with the trans-Golgi network and endoplasmic reticulum. ESEs can mature into late-sorting endosomes, which eventually generate MVBs, also known as multivesicular endosomes. The inward invagination of the endosomal limiting membrane results in the formation of MVBs containing several ILVs that are ultimately secreted as exosomes with a size range of approximately 40 to 160 nm in diameter through MVB fusion to the plasma membrane and exocytosis [5] (Figure 1). As Kalluri et al. [4] described, the MVBs degradation can occur via two pathways: lysosomes or autophagosome-assisted degradation, or plasma membrane resorption with exosome formation (ILVs release).

Exosomes exhibit heterogeneity that could be attributed to their size, content, functional impact on recipient cells, and cellular origin [7]. Additionally, the microenvironment and biology of cells may affect the content of exosomes and their biological markers. Exosomes can carry membrane proteins, cytosolic and nuclear proteins, extracellular matrix proteins, metabolites, and nucleic acids, such as mRNA, noncoding RNA species, and DNA [1,10,11,12]. Although exosomal cargo analyses require large pools of purified exosomes, the abundance of a given cargo in exosomes can differ, as shown by miRNA exosomal cargo [13].

Also, the functional heterogeneity that the exosomes show is associated with the recipient cells due to their variable expression of cell surface receptors, resulting in different effects, such as cell survival, apoptosis, and immunomodulation, among others, in different target cell types. The source organ or tissue of exosomes, including whether they originate from cancer cells [14], could also affect their properties, such as tropism to certain organs and uptake by specific cell types. Therefore, the complexity and heterogeneity of exosomes could be the result of a combination of these features.

Traumatic brain injury (TBI) is a major cause of death and disability worldwide, with no clinically successful treatment available to date [15,16]. TBI is commonly known as the sequela of head injuries, causing a sudden and rapid movement of the brain inside the skull [17,18]. These events could occur in sports or various types of professions, as well as in violence, accidents, or falls [19]. Penetrating or non-penetrating brain trauma could be the result of focal penetration of foreign objects, blunt force, jolts, jerks, bumps, or blasts [20]. The pathological processes underlying TBI are direct or indirect consequences of the brain trauma and include tissue deformation and focal damage (skull, meninges, or brain), as well as blood-brain barrier and brain blood flow disturbance, molecular imbalance associated with necrosis and apoptosis, oxidative stress, and inflammation [20]. In addition, the primary consequences of the brain injury further trigger a cascade of molecular events that lead to exacerbated responses to the initial damage, or neurological damage causing varied types of physical or mental disability [20,21]. The latest reports cited global estimates of 69 million TBI cases, with more than 27 million new cases annually [22,23]. There are a few reports of TBI incidence in Romania, estimating it at approximately 300 cases per 100,000 inhabitants per year [24,25], while in the UK, approximately 1.4 million patients reach the emergency departments for head injuries [26]. The treatment for TBI usually consists of rest, pain relievers, anti-seizure medication, diuretics, coma-inducing agents, or surgery, depending on the severity of the brain damage [27,28]. In addition, it implies a fastidious rehabilitation process, while some patients exhibit irrecuperable post-traumatic disabilities [22]. In this context, therapies that could prevent or limit the brain injury’s secondary consequences could be of much help in treating severe TBI patients.

Mesenchymal stem cell (MSC)-based therapy has been identified as a promising approach for TBI treatment, but the precise mechanisms underlying its therapeutic effects are still not fully understood [29]. Due to their ability to transfer proteins, lipids, mRNA, and miRNA between cells and promote cell-to-cell communication, exosomes are regarded as potential therapeutic agents in TBI. Moreover, as exosomes can interact with brain parenchymal cells and the neurogenic niche, leading to neurogenesis and brain remodeling, the implication of exosomes in TBI treatment can be beneficial for cognitive repair. In addition to being potent candidate biomarkers for mild TBIs [30,31], miRNAs were reported as therapeutic targets for neurorehabilitation, decreasing brain edema, and inflammation [32]. The family of miR-124 is composed of the most abundant microRNAs within the brain [33]. In animal models of TBI, miR124-3p showed significant anti-inflammatory and neuroregenerative potential [34], thus proving a worthy candidate for treating the long-term effects of TBI. Moreover, Zhu et al. [35] recently reported correlations between miR-124-3p and cognitive impairments associated with PTSD.

Thus, we aimed to review the most recent studies investigating the role of miR-124-3p-containing exosomes in TBI treatment and highlight the advantages of this approach. We also aimed to discuss the potential limitations of exosome therapy and identify areas for further research. Understanding the mechanisms underlying exosome-mediated TBI repair could provide new insights into the development of effective therapeutic strategies for TBI.

## 2. Materials and Methods

### 2.1. Literature Search

A comprehensive literature search was conducted using electronic databases such as PubMed, Scopus, and the Web of Science. The keywords used were “traumatic brain injury”, “concussion”, “post-concussion syndrome”, “exosomes”, “microvesicles”, “extracellular vesicles”, “mesenchymal stem cells”, and “miR-124-3p”. The search was limited to articles published in English between 2015 and 2023.

### 2.2. Inclusion and Exclusion Criteria

Studies were included if they were related to the role of exosomes in the treatment of TBI. Only articles published in peer-reviewed journals were included in this review. Studies that focused on exosomes role in other diseases or injuries were excluded.

### 2.3. Data Extraction and Analysis

Two reviewers independently extracted data from the selected studies. The extracted data included the author, year of publication, study design, sample size, type of exosomes used, administration route, outcomes measured, and conclusions. The data were analyzed qualitatively.

### 2.4. Data Synthesis

The extracted data were synthesized and presented in a narrative format. The findings were discussed in the context of their potential clinical applications and future research directions.

## 3. Results

According to the latest research on the molecular mechanisms in which exosomes are implicated after TBI, they appear to promote functional recovery and reduce neuronal apoptosis by modulating microglia and astrocyte activation during brain injury. Recent studies presented a wide range of TBI animal models on which the effects of exosome administration were tested. In addition to in vivo experiments, in vitro studies had the purpose of describing some of the molecular pathways through which the beneficial effects of exosomes are obtained (Figure 2).

In this study, we focused on mesenchymal stem cell-derived exosomes, as they could be crucial in tissue regeneration and inflammation [36,37,38,39,40,41,42,43,44,45,46,47,48,49]. Most of the studies we included in this review evaluated the effects of MSC-Exo that were obtained from umbilical cord stem cells [36,38,40,47,48], bone marrow [42,43,44,45,46,49], or adipose tissues [37], all of which are the best-known sources of multipotent stem cells. The methods through which exosomes are obtained from MSCs include multiple-step MSC cultivation and exosome isolation. Firstly, the MSCs (disregarding their source) are cultivated on specific cell culture media, and successive passages are performed until the exosome separation process [36,37,38,39,40,41,42,43,44,45,46,47,48,49]. In addition to the natural potential of MSCs to secrete exosomes, some studies have shown that MSC cultures are stimulated to produce exosomes with better biocompatibility and functions when exposed to certain types of stress (hypoxia) [38]. After obtaining the primary cultures of MSCs, the cells are harvested and washed, and the exosome separation process begins. There are multiple isolation methods, including commercially available kits [50,51]. However, in most of the selected studies, the MSC-Exo were mainly administered as extracts of MSCs obtained via ultracentrifugation at 100,000× *g* with or without sucrose gradient.

Xian et al. investigated the therapeutic effects of mesenchymal stem cell-derived exosomes (MSC-Exo) on inflammation-induced alterations in astrocytes, which play a key role in maintaining brain homeostasis and responding to injury [36]. It was shown that reactive astrogliosis and inflammatory responses can be attenuated when MSC-Exo is incorporated into hippocampal astrocytes both in vitro and in vivo. In this way, the amelioration of learning and memory impairments in mice with inflammation-induced astrocytic activation was obtained. They suggested that the Nrf2-NF-κB signaling pathway is involved in regulating astrocyte activation in mice and that MSC-Exo may be a promising nanotherapeutic agent for the treatment of neurological diseases with hippocampal astrocyte alterations.

In a weight-drop-induced TBI rat model, Chen et al. obtained functional recovery after injecting human adipose mesenchymal stem cell-derived exosomes intracerebroventricularly [37]. Moreover, the study showed that the beneficial effects of these exosomes were correlated with suppressed neuroinflammation, reduced neuronal apoptosis, and increased neurogenesis [37] and proposed that the mechanisms through which the exosomes act are related to entering microglia/macrophages and suppressing their activation during brain injury.

Liu et al. also explored the potential of mesenchymal stem cell-derived exosomes in TBI treatment [38]. They showed that 3D-printed collagen/silk fibroin/hypoxia-pretreated human umbilical cord mesenchymal stem cell-derived exosome scaffolds exhibit favorable physical properties that suggest increased biocompatibility and biodegradability in a TBI beagle dog model. Their effects include promoting neuroregeneration and angiogenesis while inhibiting nerve cell apoptosis and proinflammatory factor expression. Furthermore, they showed that hypoxia-induced exosomes could offer superior biocompatibility and neuroregeneration ability, suggesting that hypoxia-driven stress could promote reparatory processes. Similarly, Li et al. [39] showed that the recovery of neurological function could be achieved by intracerebroventricular administration of umbilical cord MSC-derived exosomes and suggested that their mechanism of action is correlated with the suppression of microglia and astrocyte responses to brain injury.

Also, Zhang et al. showed that exosome treatments improved sensorimotor and cognitive function, reduced hippocampal neuronal cell loss, promoted angiogenesis and neurogenesis, and reduced neuroinflammation in a rat model of TBI [40] and set the scene for a preliminary timeline for the efficiency of exosome treatment following TBI by suggesting that the administration of exosomal products at 1 day post-TBI was more efficient in offering functional and histological outcomes as compared to the other two delayed treatments.

Ghosh et al. discussed the potential of exosome therapies for neural injuries and highlighted their advantages [41] by stating that human stem cells and MSC-Exo could be a promising approach for neuronal injury healing.

Furthermore, exosome treatment attenuated the severity of neurologic injury and allowed for faster neurologic recovery in a clinically realistic large animal model of TBI and hemorrhagic shock. MSC-Exo were shown to retain some of the characteristics of their parent MSCs, such as immune system modulation, regulation of neurite outgrowth, promotion of angiogenesis, and the ability to repair damaged tissue. In a brain transcriptome study, Williams et al. [42] showed that a single dose of MSC-Exo induces transcriptomic changes that are suggestive of neuroprotection.

Han et al. investigated the effect of administration of multipotent MSC-Exo on functional recovery, neurovascular remodeling, and neurogenesis in a rat model of intracerebral hemorrhage and suggested that MSC-Exo effectively improves functional recovery, possibly by promoting endogenous angiogenesis and neurogenesis [43].

Liu et al. investigated the potential of incorporating bone marrow MSC-Exo into hyaluronan-collagen hydrogel to achieve both mimicking of brain matrix and steady release of exosomes, thus realizing TBI repair [44]. Their results demonstrated that the combination of exosomes and hydrogel effectively induced angiogenesis and neurogenesis, promoting axonal regeneration, remyelination, synapse formation, and even brain structural remodeling, leading to the neurological functional recovery of TBI.

Lu et al. investigated the effect of systemic administration of bone mesenchymal stem cell-derived extracellular vesicles on the loss of motor function after spinal cord injury and examined the potential mechanisms underlying their effects [45]. Their results showed that these extracellular vesicles could reduce brain cell death, enhance neuronal survival and regeneration, and improve motor function when administered 30 min after the spinal cord injury in the tail vein of the rats. This study suggested that extracellular vesicles could reduce pericyte migration via downregulation of NF-κB p65 signaling and further modulate the permeability of the blood-spinal cord barrier.

Zhang et al. investigated the protective effects of human umbilical cord MSC-derived exosomes in both in vivo and in vitro TBI models [46]. Their results revealed that exosome treatment improved neurological function, decreased cerebral edema, attenuated brain lesions after TBI, and suppressed TBI-induced cell death, apoptosis, pyroptosis, and ferroptosis. Additionally, the exosomes activated the PINK1/Parkin pathway-mediated mitophagy after TBI.

Cui et al. isolated exosomes derived from umbilical cord mesenchymal stem cells and intraventricularly injected them into a rat model of TBI. The results demonstrated that the studied exosomes promoted functional recovery and reduced neuronal apoptosis in TBI rats [47]. Furthermore, the study found that umbilical cord mesenchymal stem cell-derived exosomes inhibited the activation of microglia and astrocytes during brain injury, which may contribute to functional recovery. However, the study failed to demonstrate the modulatory effects of this treatment on the inflammatory factors that were evaluated in the rats’ plasma. Overall, the findings suggest that umbilical cord mesenchymal stem cell-derived exosomes may have therapeutic potential for TBI by inhibiting microglia and astrocyte activation and promoting functional recovery. These results provide a basis for the development of new therapeutic strategies for central nervous system diseases.

The administration of MSC-Exo within one hour after TBI and hemorrhagic shock has been shown to provide neuroprotection in animal models [48]. However, the underlying mechanisms responsible for the neuroprotective effects are not completely understood. To further investigate this treatment strategy, this study aimed to analyze the transcriptomic changes in the brain associated with the administration of MSC-Exo. A Yorkshire swine weighing 40–45 kg was subjected to a severe TBI and hemorrhagic shock. After one hour of shock, animals were randomly assigned to receive either lactated Ringer’s or exosomes suspended in lactated Ringer’s. Brain swelling and lesion size were assessed after 6 h of observation, and peri-injured brain tissue was processed for RNA sequencing. Results from high-throughput RNA sequencing data analysis were compared between the control and experimental groups. In this experimental context, it was demonstrated that a single dose of exosomes could particularly modulate the expression of some genes associated with blood-brain barrier stability, neurogenesis, neuronal development, synaptogenesis, and neuroplasticity, as well as genes associated with stroke, neuroinflammation, neuroepithelial cell proliferation, and non-neuronal cell proliferation (reactive gliosis). However, further studies are needed to determine the potential of exosomes as a treatment for TBI in humans and the traceability of effects on the transcriptome correlated with neuroprotection, as seen in the animal models.

Regarding the efficiency of exosome treatment, Williams et al. showed that even a single-dose administration could attenuate the effects of neurological injuries by reducing the swelling of the brain lesion and modulating inflammation and apoptosis in a 7-day survival model using a swine animal model of TBI [49]. Moreover, it was found that this single dose of exosomes could promote neural plasticity that could last as long as 7 days. Specifically, the exosome-treated animals had significantly lower neurologic severity scores, faster neurologic recovery, smaller brain lesion sizes, lower levels of inflammatory markers, and lower levels of apoptotic markers. They also had higher levels of the mediator of neural plasticity, brain-derived neurotrophic factor.

Ge et al. investigated the role of miRNAs in microglial exosomes in regulating post-traumatic neurodegeneration [52]. The study demonstrated that the level of miR-124-3p in microglial exosomes from the injured brain was significantly altered in the acute, sub-acute, and chronic phases after repeated mild TBI. In this way, in a reoccurring scratch-injury model, they showed that miR-124-3p-upregulated microglial exosomes could significantly decrease the activity of the pathways associated with neurodegeneration. They proposed that this effect was obtained due to the potential of microglial exosomes to target Rela, an inhibitory transcription factor that is implicated in β-amyloid proteolytic breakdown and thereby a potential inhibitor of β-amyloid pathway abnormalities. In this mouse model, it was observed that the intravenous administration of microglial exosomes could have significant effects within the brain, meaning that they can pass through the blood-brain barrier—this being an important feature of treatments that target the central nervous system. Additionally, miR-124-3p within the exosomes was transferred into hippocampal neurons and alleviated neurodegeneration by targeting the Rela/ApoE signaling pathway, suggesting that these exosomes could also have cognitive effects following repeated mild TBIs.

Huang et al. also used animal models of rodents and repetitive TBI-like injuries to study neuronal inflammation modulation using microglial exosomes and miRNA activity against injured neurons [35]. In this way, the injured brains of the mice were collected during the acute and chronic phases of TBI, and an extract was used on in vitro-cultured microglia. The analysis of the microglial exosome-derived miRNA showed a significant increase in miR-124-3p levels that led to neuroinflammation inhibition, possibly by promoting the anti-inflamed M2 polarization in microglia and by inhibiting mTOR signaling and PDE4B gene expression. Moreover, exosomal miR-124-3p contributed to neurite outgrowth, characterized by an increase in the number of neurite branches and total neurite length and a decrease in the expression of neurodegenerative proteins (Aβ-peptide and p-Tau). Furthermore, neuroprotective and anti-inflammatory properties were shown in repetitive TBI designs. These results suggest that miR-124-3p-containing exosomes could exhibit significant neuroregenerative and anti-inflammatory properties that could provide an innovative therapeutic approach for brain injury and other neurological disorders.

In their study, Zhao et al. investigated the effects of miR-124-3p on BMVEC function to determine the mechanistic basis of its activity [53]. The bEnd.3 cell scratch wound model was used to simulate TBI-associated endothelial cell injury, and Lipofec-tamine3000 was employed to overexpress miR-124-3p in endothelial cells. The upregulation of miR-124-3p was found to suppress bEnd.3 cell apoptotic death following in vitro scratch injury while promoting the upregulation of the tight junction proteins ZO-1 and occludin in these cells, thereby reducing the degree of BBB leakage. Thus, due to its ability to modulate mTOR signaling and autophagy, it is thought that miR-124-3p exerts protective effects against apoptosis. Once again, it was shown that miR-124-3p could modulate mTOR signaling and autophagy, thus contributing to their protective properties (Table 1).

## 4. Discussion

A recent study [54] pointed out that microRNAs that are found within the exosomes could be potent molecular makers used in central nervous system disease diagnosis as their specific activity variation could provide additional information regarding the pathological changes in the brain tissues. Furthermore, they emphasized their role in neural regeneration and their potential to be exogenously obtained and transferred to exosomal vesicles, making them excellent therapeutic targets. Despite being a major public health concern worldwide [22,23,24,25,26], the lack of effective treatments increases the burden of TBI on patients and caregivers. The current treatment approaches for TBI effects are based on symptoms relief, rest, and, in more severe cases, on the modulation of brain inflammatory processes, as well as brain edema, hemorrhages, or surgery. These treatment alternatives offer a good prognosis for patients’ survival and, in combination with physical and neurological rehabilitation, show significant improvement in the quality of life and physiological functioning of the patients [27,28]. However, there are some TBI molecular impairments that could cause permanent disability for the sufferers [26]. In this context, ways to prevent the development of permanent damage or to induce tissue regeneration could be an important step in TBI management and treatment.

The potential therapeutic effects of MSC-derived and microglial exosomes have been widely studied in recent years, with promising results [55,56,57,58,59]. The exosomes potential to be used as therapeutic agents is given by their ability to carry proteins, lipids, mRNA, and miRNA and to be implicated in the communication between cells [60,61]. Studies have shown that MSC-derived exosomes can contribute to the therapeutic effects of MSCs in TBI and hemorrhagic shock models, promoting neurogenesis and brain remodeling [43,49,50]. Administration of a single dose of exosomes induces transcriptomic changes suggestive of neuroprotection, which can reduce genes associated with stroke, neuroinflammation, and non-neuronal cell proliferation, contributing to reactive gliosis. As previously emphasized by the findings regarding the TBI pathophysiological mechanisms, an important pathway that contributes to TBI’s permanent and debilitating damage is selective hippocampal damage. The hippocampal vulnerability to TBI-driven damage was previously described by Royo et al. [62], who showed that ipsilateral hippocampal areas are prone to manifest neuronal loss as compared to the hippocampal area contralateral to the TBI damage. It was proposed that selective hippocampal damage due to TBI is correlated with hypoxia, glutamate toxicity, hyperexcitability, and neuroinflammation [63]. In this context, Chen et al. [37] reported that intracerebroventricularly injected human adipose MSC-Exo could significantly improve hippocampal neurogenesis in a weight-drop-induced TBI rat model, thus specifically addressing this TBI effect. Recent data added even more information about the correlation between the mechanisms behind TBI by proposing possible means by which secondary injuries due to TBI are produced. Thus, besides the primary injury that could occur to the brain-blood barrier due to direct shearing following the brain injury, the secondary injury to this barrier could be the result of astrocytic dysfunction and inflammation [64]. Furthermore, the molecular cascade that leads to the neuroinflammatory response further enhances even more the inflammatory status within the affected brain tissues, which is also an important player in the brain-blood barrier disruption process as well as in the occurrence of oxidative stress [65]. Once oxidative stress is present, further impairments result regarding the processes associated with inflammation, astrocyte and microglia activation, and mitochondrial damage [65]. In this context, it was shown that mitochondria-associated microRNA expression could be altered in the brains of animal models of TBI, suggesting that hippocampal mitochondria respond differently to stressors as compared to cortical mitochondria [66].

On the other hand, some studies suggested that selective hippocampal damage due to TBI could be mediated by gap junctions and hemichannels between the glial cells. In this way, both Chen et al. [67] and Mayorquin et al. [68] described a mechanistic association between connexins and pannexins and exosome release in the hippocampal area. In a rat model of TBI, a specific connexin phosphorylation, connexin 43, leads to tissular damage propagation and glial exosome release in the ipsilateral hippocampus. In the context of miR-124-3p treatment in TBI, it was noted that in some brain tumoral cells, this microRNA gains antiproliferative abilities and travels through the connexin 43 channels [69,70]. Similarly, pannexin-1 channel inhibition was correlated with decreased neuroinflammation following TBI and hypoxic ischemia [71] and could be an important target in TBI treatments. In this context, MSC-derived exosomes effectively improve functional recovery in a rat model of intracerebral hemorrhage by promoting endogenous angiogenesis and neurogenesis. Another study demonstrated that incorporation of MSC-derived exosomes into hyaluronan-collagen hydrogel can achieve both mimicking of the brain matrix and steady release of exosomes, resulting in TBI repair [44,45,47]. Similarly, microglial exosomes have been found to have a potential therapeutic role in TBI [51,52]. They can help modulate inflammation, reduce oxidative stress, and promote neurogenesis.

Microglia are resident immune cells in the central nervous system that play an important role in the pathophysiology of TBI. Following a brain injury event, the microglia are the first immunological actors in the central nervous system that respond to injury by changing their morphology to an ameboid shape, exhibiting phagocytic activity, and releasing inflammatory agents in the surroundings [72]. The dual activity of microglia in inflammation was demonstrated by the description of the M1 and M2 phenotypes that exhibit opposite inflammatory activities, anti-inflammatory and pro-inflammatory, respectively [73]. Thus, M2 microglia can release exosomes containing cytokines, chemokines, and microRNAs, which can interact with other central nervous system cells to promote neuroprotection and repair. Of the microRNAs implicated in this process, the miR-124 family of microRNAs has been shown to promote the polarization of microglia into the M2 phenotype and hippocampal neurogenesis [74], as discussed before. The molecular mechanism by which miR-124 modulated M2 polarization of the microglia was by inhibiting TLR4 signaling pathway components expression, which is also implicated in hippocampal neuron loss [75]. This could be an important target for exosome-based therapeutic approaches in TBI treatment.

Regarding the potential of miR-124-3p to modulate cellular and functional recovery following TBI, there are several studies that demonstrate that microglial exosomes that contain increased levels of miR-124-3p are implicated in the reparatory processes within the TBI-affected brains [52]. Moreover, miR-124-3p was seen to be implicated in β-amyloid metabolism within the brain, thus being able to prevent the Alzheimer’s disease-like neurodegenerative-related processes seen in the long-term effects of TBI [51]. However, its main contribution remains neuronal inflammation and neurite outgrowth when transported into the neurons. In addition, several studies described their activity within the brain microvascular system by promoting mTOR signaling and subsequent autophagy, as well as improving the cerebral microvascular endothelial barrier [35].

The mechanisms of action of miR-124-3p include the potential to bind to untranslated regions of mRNAs that code for important pathway modulators, such as the mTOR pathway [52], the neurotrophin signaling pathway, and the cAMP signaling pathway [51,76]. Voukila [77] also described the binding site targets for miR-124-3p as post-transcriptional switches involved in non-neuronal splicing, activated by miR-12-3p downregulation, which could be caused by target RNA-directed miRNA degradation.

Overall, the studies suggested that exosomes derived from MSCs or other stem cells have significant advantages and could be considered a potential therapeutic approach for TBI and SCI. The main advantages consist in their effective transport and their multiple properties in neuroprotection, endogenous angiogenesis, and neurogenesis, as well as in the inhibition of apoptosis, pyroptosis, and ferroptosis.

Considering the strategies and outcomes of the discussed studies, there are also some limitations to mention. For instance, the heterogeneity of exosomes can be a significant challenge in their therapeutic applications. Exosomes can have different sizes, contents, and functional impacts on recipient cells, which can result in different therapeutic effects. In addition, the isolation and purification methods of exosomes can affect their quality and quantity, and the potential for contamination with other extracellular vehicles is always present.

Therefore, future perspectives in research could focus on the isolation and characterization methods of exosomes to ensure their therapeutic efficacy and safety in clinical applications. Moreover, as the mechanisms by which the exosome functions are performed are not fully understood nor their contents fully described, the efficacy and safety of these therapies in human clinical trials should be preceded by characterization and mechanistic studies. Moreover, miR-124-3p might not be the only microRNA contained by the microglial exosomes that could exert beneficial effects in TBI, not only within the brain cells but also in other affected tissues. Thus, further studies could focus on describing the microRNA content of microglial exosomes and specifically studying their effects on brain and bone tissues, as also seen for miR-21-5p [78,79] or miR-195, which targets NOD-like receptor X1 that interacts with the NF-κB signaling pathway by inhibiting it and preventing the inflammatory response during secondary brain injury and thus promoting structural and functional recovery after TBI [80].

## 5. Conclusions

The therapeutic potential of mesenchymal and microglial exosomes in TBI and PCS is promising. They can promote neuroprotection, repair, and regeneration by modulating inflammation, reducing oxidative stress, and promoting neurogenesis. Further research is needed to optimize their isolation and characterization methods and to elucidate their mechanisms of action, but the potential of exosomes as a novel therapeutic strategy for TBI and PCS is significant.

## Figures and Tables

**Figure 1 life-13-01924-f001:**
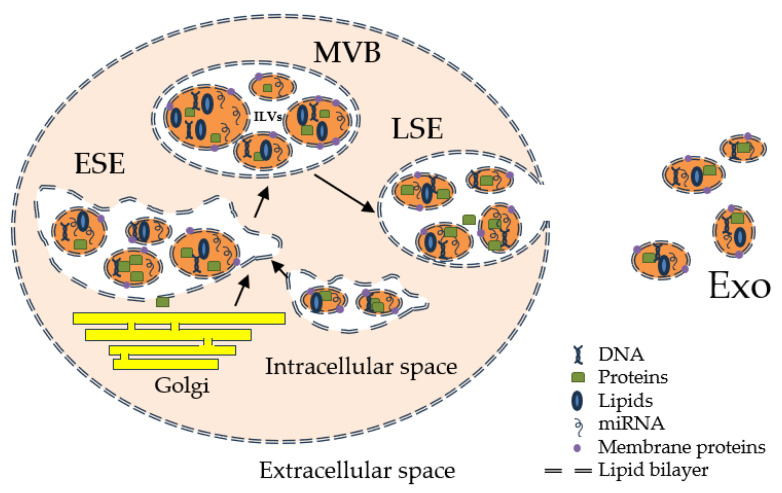
Exosome biogenesis and release.

**Figure 2 life-13-01924-f002:**
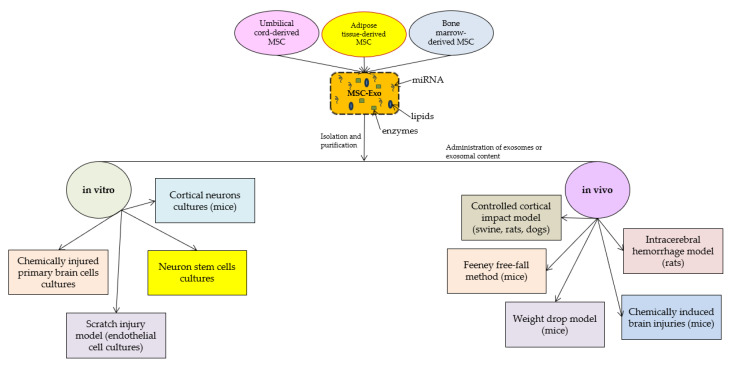
Sources of exosomes and models in which they proved their efficiency.

**Table 1 life-13-01924-t001:** Summarized presentation of the studies regarding the effects of MSC-Exo in TBI models.

Study	Objective	Models and Administration	Key Findings
[36]	The effects of MSC-Exo on inflammation-induced astrocytic alterations	in vitro: hippocampal astrocytic inflammation (lipopolysaccharide administration to primary cell cultures); in vivo: inflammation-induced astrocytic activation and status epilepticus (in mice, pilocarpine administration); MSC-Exo was obtained from human umbilical cord tissues in specific culture conditions and using ultracentifugation methods for separation.	MSC-Exo were incorporated into hippocampal astrocytes; they attenuated reactive astrogliosis and inflammatory responses; they ameliorated learning and memory impairments.
[37]	The effects of MSC-Exo on functional recovery, neuroinflammation, neuronal apoptosis, and neurogenesis	in vivo: weight-drop TBI rat model; MSC-Exo were obtained from human adipose tissues and administered intracerebroventricularly by microinjection.	MSC-Exo promoted functional recovery, suppressed neuroinflammation, reduced neuronal apoptosis, and increased neurogenesis; specifically entered microglia/macrophages and suppressed their activation during brain injury.
[38]	The regenerative potential of MSC-derived exosomes	in vivo: TBI canine model (Beagle dogs receiving small shocks using an electronic cortical contusion impactor through a hole in the skull); MSC-Exo were obtained from hypoxia-treated human umbilical cords and were used for a 3D-printed collagen/silk fibroin/exosomes scaffold.	MCS-Exo significantly promoted neuroregeneration and angiogenesis; they inhibited nerve cell apoptosis and proinflammatory factor expression.
[40]	The effects of MSC-Exo in relation to doses and times of administration	in vivo: unilateral moderate cortical concussion rat models. MSC-Exo were obtained from human umbilical cord tissues and separated using ultra-centrifugation in the sucrose gradient method; MSC-Exo were administered in the tail vein of the rats at different time points after brain injury.	MSC-Exo treatment improved sensorimotor and cognitive function, reduced hippocampal neuronal cell loss, promoted angiogenesis and neurogenesis, and reduced neuroinflammation; One day post-TBI administration provided significantly greater improvements in functional and histological outcomes as compared to delayed treatments.
[42]	The effects of bone marrow-derived MSC-Exo on TBI and hemorrhagic shock in animal models	in vivo: TBI and hemorrhagic shock swine models (computer-controlled cortical impact device); MSC-Exo were separated from a single human bone marrow donor and administered intravenously at 9 h, 1, 5, 9, and 13 days post-injury.	A single administration of MSC-Exo resulted in attenuated brain injury effects and better functional neurological recovery; MSC-Exo-mediated central nervous system and peripheral immune response.
[43]	The effects of multipotent MSC-Exo on functional recovery, neurovascular remodeling, and neurogenesis	in vivo: Wistar rat model of intracerebral hemorrhage (blood injection into the striatum); MSC-Exo were obtained from bone marrow of Wistar rats via ExoQuick exosome isolation method after culture standard procedures; MSC-Exo were administered intravenously 24 h after the injury.	significant improvement in functional neurological recovery (spatial learning and motor functions) after MSC-Exo treatment; possible modulation of endogenous angiogenesis and neurogenesis.
[44]	The efficiency of hyaluronan-collagen hydrogel incorporating bone marrow MSC-Exo in TBI treatment	in vivo: Sprague-Dawley rat model of direct brain injury; MSC-Exo were obtained from the bone marrow of Sprague-Dawley rats via ultracentrifugation; The MSC-Exo-hydrogel complex was administered to rats after brain injury.	MSC-Exo-induced angiogenesis and neurogenesis; suggestive brain structural remodeling.
[45]	The effects of MSC-Exo treatment on spinal cord injury	in vivo: rat model of spinal cord injury (contusive injury, spinal cord impactor); MSC-Exo were isolated from rat bone marrow via ultracentrifugation and administered intravenously 1 day post-injury.	promotion of regeneration and neuronal survival, reduced brain cell death, and improved motor functions; NF-κB p65 signaling in pericytes—a possible action pathway of MSC-Exo.
[46]	The neuroprotective effects of human umbilical cord MSC-Exo	in vitro: mouse cortical neurons cultures exposed to mechanical stretch injury. in vivo: mouse models of TBI (controlled cortical impact). human umbilical cord MSC-Exo were obtained from expanded cultures via ultracentrifugation.	improvement of neurological function and cerebral edema reduction, contributing to brain regeneration; suppressed apoptosis, pyroptosis, and ferroptosis; modulated PINK1/Parkin pathway-mediated mitophagy.
[47]	The reparatory mechanism to which MSC-Exo contributes	in vivo: rat model of Feeney’s free-fall method for brain injury; MSC-Exo were isolated from the third to fifth generation of umbilical cord stem cell cultures via ultracentrifugation.	reducing neuronal apoptosis and promoting functional recovery; inhibition of microglia and astrocytes.
[48]	The neuroreparatory molecular mechanisms that the MSC-Exo is contributing to	in vivo: swine model of severe TBI and hemorrhagic shock; bone marrow-derived MSC-Exo were administered intravenously 1 h after the injury.	MSC-Exo promoted the expression of genes associated with neurogenesis, neuronal development, synaptogenesis, and neuroplasticity; MSC-Exo inhibited the expression of genes associated with stroke, neuroinflammation, neuroepithelial cell proliferation, and non-neuronal cell proliferation.
[49]	The impact of early single-dose exosome treatment in a 7-day survival model	in vivo: swine model of severe TBI and hemorrhagic shock; bone marrow-derived MSC-Exo were administered intravenously 1 h after the injury.	A single-dose administration of MCS-Exo improved survival after TBI; decreased brain lesion size; inhibition of inflammation and apoptosis; promotion of neural plasticity.
[51]	The role of miRNAs in regulating post-traumatic neurodegeneration	in vivo: mouse TBI models (close impact injury); Exosomes were isolated from the brains of the animals after incubation and ultracentrifugation.	miR-124-3p modulated neurodegeneration and inhibited β-amyloid abnormalities; miR-124-3p could pass through the blood-brain barrier; miR-124-3p improved the cognitive outcome after repetitive mild TBI.
[35]	To explore the regulatory mechanism of microglial exosomes on neuronal inflammation in TBI by investigating the impact of microglial exosomal miRNAs on injured neurons	in vivo: mouse model of controlled cortical impact; in vitro: scratch injury model of cultured neurons; BV2 microglia were treated with brain extracts, and exosomes were isolated via ultracentrifugation; miR-124-3p mimics were transfected into microglial cultures.	miR-124-3p inhibited neuronal inflammation and contributed to neurite outgrowth.
[53]	The effects of miR-124-3p on brain microvascular endothelial cell function and their molecular mechanisms	in vitro: a specific cell scratch wound model for endothelial cell injury; miR-124-3p mimic was transfected into cell cultures.	miR-124-3p overexpression prevented apoptosis and reduced blood-brain barrier leakage; miR-124-3p exerts protective properties against damage by promoting autophagy and suppressing mTOR signaling.

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
