# Peer review of "The Role of Microglial Exosomes and miR-124-3p in Neuroinflammation and Neuronal Repair after Traumatic Brain Injury"

_life, 2023, doi:10.3390/life13091924_

Round 1

Reviewer 1 Report

The manuscript focuses on applying exosomes and MiR-124-3p in traumatic brain injury. After a thorough review, some concerns need to be addressed.

Comment 1: The review is concise, with only 42 references from a specific period from 2015-2024. If we search exosomes for traumatic brain injury in PubMed, it shows more than 100 reports. Adding more references to the manuscript is suggested to make it informative and exciting to the target audience.

Comments 2: The review looks like a bird eye view; there should be detailed explanations of the advantages, limitations, and future perspectives of using exosomes and MiR-124-3p in TBI. It would attract the audience.

Comment 3: The quality of the graphical abstract and Figure 1 should be improved.

The manuscript is readable, language is fine.

Author Response

“The manuscript focuses on applying exosomes and MiR-124-3p in traumatic brain injury. After a thorough review, some concerns need to be addressed.

Comment 1: The review is concise, with only 42 references from a specific period from 2015-2024. If we search exosomes for traumatic brain injury in PubMed, it shows more than 100 reports. Adding more references to the manuscript is suggested to make it informative and exciting to the target audience.”

Response: We revised the manuscript according to all three reviewers’ comments and the references number improved as many additional information was added.

“Comments 2: The review looks like a bird eye view; there should be detailed explanations of the advantages, limitations, and future perspectives of using exosomes and MiR-124-3p in TBI. It would attract the audience.”

Response: We revised the ways in which the discussions were presented and tried to add some advantages, limitations, and future perspectives. Also, we added in the introduction several information on the action of miR-124-3p and the reason it was considered in this study.

“Comment 3: The quality of the graphical abstract and Figure 1 should be improved.”

Response: We revised the graphical abstract and Figure 1 for quality and provided separate files for each of them. 

Reviewer 2 Report

The review is focused upon a specific subject, such as the role of exosomes in traumatic brain injury. Therefore, it is clear that selection of papers to be discussed must be flexible, but also precise at the same time.

I think that authors managed to choose papers to discuss well. However, some small amendments are necessary to make the subject of exosomes and traumatic brain injury more understandable to readers.

- In my opinion short explanation what is a traumatic brain injury, together with a short, concise description of the pathological process is necessary to understand different experimental models described in the paper. Figure 1 is good but not enough to guide through the paper.

- a figure showing the mechanism of exosome biogenesis and release from cells would be a good addition to introduction.

- Exo isolation and enrichment methods should be shortly described since different experimental settings are cited in the paper.

- I am not convinced that only mir 124-3p is crucial for recovery from TBI. Can authors comment this?

Small grammatical and vocabulary errors, for example:

Line 86: this/thus

Line 128: amelioration of ameliorated

Table 1:

The phrases are too long, shortening would help to understand the cited paper.

Author Response

“The review is focused upon a specific subject, such as the role of exosomes in traumatic brain injury. Therefore, it is clear that selection of papers to be discussed must be flexible, but also precise at the same time.

I think that authors managed to choose papers to discuss well. However, some small amendments are necessary to make the subject of exosomes and traumatic brain injury more understandable to readers.

- In my opinion short explanation what is a traumatic brain injury, together with a short, concise description of the pathological process is necessary to understand different experimental models described in the paper. Figure 1 is good but not enough to guide through the paper.”

Response: We newly added in the introductory section some aspects regarding TBI. Also, we revised Figure 1 to be more comprehensive.

“- a figure showing the mechanism of exosome biogenesis and release from cells would be a good addition to introduction.”

Response: We added a figure regarding the mechanisms of exosomes biogenesis and release.

“- Exo isolation and enrichment methods should be shortly described since different experimental settings are cited in the paper.”

Response: We newly explained these aspects within the beginning of the results section.

“- I am not convinced that only mir 124-3p is crucial for recovery from TBI. Can authors comment this?”

Response: It is true that mir124-3p might not be the only microglial exosome that could help in TBI recovery, due to the relevant data presented by the recent studies (as added in the introduction, together with other reasons we decided to focus on this one). However, we mentioned that further perspectives would be to focus on other microglial exosomes as well.

“Comments on the Quality of English Language

Small grammatical and vocabulary errors, for example:

Line 86: this/thus

Line 128: amelioration of ameliorated

Table 1:

The phrases are too long, shortening would help to understand the cited paper.”

Response: English language quality was revised, as suggested by the reviewer. Table 1 was revised.

Reviewer 3 Report

The current review focuses on TBI as a serious issue for global public health and is a one of the leading cause of death and long-term impairment for people of all ages. Importantly, the authors have highlighted exosomal miRNAs to improve the prognosis of brain injury. The review is well designed and executed. The studies inclusion is less and must be focused on. Additionally, there are certain aspects which are untouched or needs further modifications. There are few recommendations:

1.      The graphical abstract is very basic. The authors are advised to reframe/work on graphical abstract to make it more interesting and ensure that the whole story fits in this one figure. Same applies for Fig. 1. The ideas can be framed in a better way. The authors can use sophisticated tools like Bio render and others to re-structure the figures.

2.      The authors can provide recent data on global burden of traumatic brain injury in the introduction section. Is there any regional data available (area-wise)?

3.      The authors can briefly include the current treatments for traumatic brain injury and how these patients need alternatives. The authors can highlight the usage of medications.

4.      Hippocampal damage including neurodegeneration and neuroinflammation after (TBI) is associated with late post-traumatic conditions. The authors can highlight the mechanism of selective hippocampal damage after TBI.

5.      What are the endogenous binding status and epigenetic modulation of miR-124-3p with target genes.

6.      The authors can highlight about the Connexin 43 hemichannel which is expressed especially in astrocytes and microglia during TBI including their involvement in spread of brain injury as their presence in hippocampal cells is known. Same applies for Pannexin channel. Their blockage can ameliorate the TBI and the link of microglial exosomes could be interesting.

7.      Few Micro-RNAs suppress the Nod-like receptor (NLRX1) function and overcome the unfavorable effects to neurons during TBI. Kindly add data on it.

8.      How the microglia change their morphology and functions after TBI. The redirection of microglial polarization to M2 phenotype can be considered as a one of the approaches to increase better results in TBI. The TLR-4 signaling pathway plays an critical role in the activation and polarization of microglia. The authors must elaborate this section.

9.      The exosomal miRNAs are not only used as biomarkers of TBI diagnosis but are promising therapeutic targets for TBI. Discuss.

10.  The discussion is very short and do not touch the major focus of the review at large. In discussion, TBI can be discussed as critical public health and socio-economic problem. With their ability to enhance functional recovery, decrease cortical lesion volume, attenuate cellular death, and regulate neuroinflammation, Exos and their miRNA cargo have been recognized as having a major role. Moreover, the “mir-124-3p” as a focused biomarker is missing in this section.

11.  How neuroinflammation contributes to the BBB disruption in the after-hypoxia insult during TBI?  The authors must highlight on the link between the Ox stress and altered expression of miRNAs.

12.  Grammar: Rephrase line 117, considered presented”

English quality is perefct. Only minor corrections.

Author Response

“The current review focuses on TBI as a serious issue for global public health and is a one of the leading cause of death and long-term impairment for people of all ages. Importantly, the authors have highlighted exosomal miRNAs to improve the prognosis of brain injury. The review is well designed and executed. The studies inclusion is less and must be focused on.”

Response: Thank you for your kind suggestions. As this aspect was also pointed out by another reviewer, the manuscript was revised for information and content and the methods of selection was improved by adding supplemental key words.  

“Additionally, there are certain aspects which are untouched or needs further modifications. There are few recommendations:

  1. The graphical abstract is very basic. The authors are advised to reframe/work on graphical abstract to make it more interesting and ensure that the whole story fits in this one figure. Same applies for Fig. 1. The ideas can be framed in a better way. The authors can use sophisticated tools like Bio render and others to re-structure the figures.”

Response: Graphical abstract and Figures were revised.

“2.      The authors can provide recent data on global burden of traumatic brain injury in the introduction section. Is there any regional data available (area-wise)?”

Response: The introductory section now benefits from some data regarding the global and regional burden of TBI.

“3.      The authors can briefly include the current treatments for traumatic brain injury and how these patients need alternatives. The authors can highlight the usage of medications.”

Response: This aspect was briefly presented in the introduction and added to the beginning of the discussion section.

“4.      Hippocampal damage including neurodegeneration and neuroinflammation after (TBI) is associated with late post-traumatic conditions. The authors can highlight the mechanism of selective hippocampal damage after TBI.”

Response: This aspect was added in the discussion section.

“5.      What are the endogenous binding status and epigenetic modulation of miR-124-3p with target genes.”

Response: We also added some discussions regarding this aspect in the discussion section.

“6.      The authors can highlight about the Connexin 43 hemichannel which is expressed especially in astrocytes and microglia during TBI including their involvement in spread of brain injury as their presence in hippocampal cells is known. Same applies for Pannexin channel. Their blockage can ameliorate the TBI and the link of microglial exosomes could be interesting.”

Response: We tried to address this issue in the discussion section.

“7.      Few Micro-RNAs suppress the Nod-like receptor (NLRX1) function and overcome the unfavorable effects to neurons during TBI. Kindly add data on it.”

Response: You can find these aspects at the end of the discussion section.

“8.      How the microglia change their morphology and functions after TBI. The redirection of microglial polarization to M2 phenotype can be considered as a one of the approaches to increase better results in TBI. The TLR-4 signaling pathway plays an critical role in the activation and polarization of microglia. The authors must elaborate this section.”

Response: Thank you. We elaborated this aspect in the discussion section.

“9.      The exosomal miRNAs are not only used as biomarkers of TBI diagnosis but are promising therapeutic targets for TBI. Discuss.”

Response: Thank you. We discussed on this issue in the first paragraph of the discussions section.

“10.  The discussion is very short and do not touch the major focus of the review at large. In discussion, TBI can be discussed as critical public health and socio-economic problem. With their ability to enhance functional recovery, decrease cortical lesion volume, attenuate cellular death, and regulate neuroinflammation, Exos and their miRNA cargo have been recognized as having a major role. Moreover, the “mir-124-3p” as a focused biomarker is missing in this section.”

Response: According to the other reviewers’ comments as well as your previous comments, the discussion was revised to include critical aspects about the issues that were mentioned.

“11.  How neuroinflammation contributes to the BBB disruption in the after-hypoxia insult during TBI?  The authors must highlight on the link between the Ox stress and altered expression of miRNAs.”

Response: Thank you. Discussion regarding this aspect could be found in the discussion section.

“12.  Grammar: Rephrase line 117, “considered presented””

Response: English language was revised throughout the manuscript.

Round 2

Reviewer 3 Report

The authors have done the required modifications and the improvements as per the suggestions. All the sections are drastically improved and the looks suitable for publication. The changes and the manuscript can be considered in the present form.